# Exposure to stereotype-relevant stories shapes children's implicit gender stereotypes

**Katharina Block** [1]*, **Antonya Marie Gonzalez**[2], **Clement J. X. Choi**[3], **Zoey C. Wong**[4], **Toni Schmader**[3], **Andrew Scott Baron**[3]

1 Department of Psychology, University of Amsterdam, Amsterdam, Noord-Holland, Netherlands, 2 Department of Psychology, Western Washington University, Bellingham, Washington, United States of America, 3 Department of Psychology, The University of British Columbia, Vancouver, British Columbia, Canada, 4 Department of Occupational Sciences and Occupational Therapy, University of Toronto, Toronto, Ontario, Canada

* k.a.e.block@uva.nl

**Data Availability Statement:** All relevant data and experiment files are available openly at: https://osf.io/2h3xu/.

## Abstract

Implicit math = male stereotypes have been found in early childhood and are linked to girls' disproportionate disengagement from math-related activities and later careers. Yet, little is known about how malleable children's automatic stereotypes are, especially in response to brief interventions. In a sample of 336 six- to eleven-year-olds, we experimentally tested whether exposure to a brief story vignette intervention with either stereotypical, neutral, or counter-stereotypical content (three conditions: math = boy vs. neutral vs. math = girl) could change implicit math-gender stereotypes. Results suggested that children's implicit math = male stereotypes were indeed responsive to brief stories that either reinforced or countered the widespread math = male stereotype. Children exposed to the counter-stereotypical stories showed significantly lower (and non-significant) stereotypes compared to children exposed to the stereotypical stories. Critically, exposure to stories that perpetuated math = male stereotypes significantly increased math-gender stereotypes over and above baseline, underscoring that implicit gender biases that are readily formed during this period in childhood and even brief exposure to stereotypical content can strengthen them. As a secondary question, we also examined whether changes in stereotypes might also lead to changes in implicit math self-concept. Evidence for effects on implicit self-concept were not statistically significant.

## Malleability of children's implicit gender stereotypes

Early engagement with math is essential for success in Science, Technology, Engineering, and Math fields (STEM) [1]. Despite gender parity in math capabilities throughout elementary and secondary school, girls begin to disengage from STEM fields in childhood [2]. This process of disengagement continues across development [3, 4], and is an important precursor of the underrepresentation of adult women in STEM fields [5]. Recent research partially attributes women's disengagement from these fields (i.e., their relatively lower STEM self-concept) to the role of prevalent automatic associations linking concepts related to STEM more to men than to women (i.e., *implicit* math-gender stereotypes) [6–8].

**Funding:** This research was supported by a Social Sciences and Humanities Research Council of Canada grant to A.S.B at the University of British Columbia (# 435-2013-0286 and #895-2016-2011). The funders had no role in study design, data collection and analysis, decision to publish, or preparation of the manuscript. No authors received salary directly from the funding agencies. The authors have declared that no competing interests exist.

**Competing interests:** The authors have declared that no competing interests exist.

Implicit gender stereotypes have negative effects on women through several avenues. For women, holding strong implicit STEM = male stereotypes is a significant predictor of lower engagement with and commitment to STEM jobs [8, 9]. Women's STEM engagement can also be indirectly affected by stereotypes when gender stereotypes held by others lead to biased evaluations [10]. For example, recent evidence suggests that implicit gender stereotypes of evaluation committees predict promotions given to women in STEM fields, especially among committees that explicitly deny the existence of gender biases [11]. Given these implications in adulthood, and the fundamental importance of early math engagement for future career paths in STEM [1], it is crucial to understand whether implicit math-gender stereotypes are malleable before these biases can influence post-secondary academic and career pursuits. To this end, the current research asked three questions; I) can a brief exposure to stereotype-relevant stories shape children's implicit math = male associations, II) does this effect vary by age, and, as a secondary question, III) does exposure to stereotype-relevant stories also shape children's self-concept directly?

## Implicit gender stereotypes and consequence in children

We conceptualize implicit cognitions as associative links between concepts that can be activated automatically [12, 13]. Importantly, implicit stereotypes do not need to be endorsed explicitly to predict behavior [8, 11–13]. By this conceptualization, implicit gender stereotypes reflect automatically activated cognitive associations between one's concept of a gender group and particular attributes (e.g., a domain like "math"). Mounting evidence suggests that implicit stereotypes about social groups, including gender, reach adult-like levels by the time children are school aged [3, 4]. Specifically, as early as elementary school, children exhibit implicit gender stereotypes associating math with boys and reading with girls [3, 4, 14]. By middle school, these stereotypes are comparable in magnitude to adult gender stereotypes [3, 4, 15].

Left unchanged, these early emerging implicit gender stereotypes have the potential to shape behavior. For example, implicit math = male stereotypes predict lower math identification and math achievement in elementary school-aged girls [3, 4]. Furthermore, when gender differences are made salient, girls with stronger implicit gender stereotypes tend to underperform on math tasks [16]. Evidence suggests that gender imbalances in STEM engagement emerge in early childhood and continue to intensify under the influence of implicit gender stereotypes. Yet the data also hint that childhood is an important window for shaping the magnitude and direction of these biases while they are initially forming [3, 4]. It is thus important to identify methods for shaping implicit gender stereotypes in children to mitigate the influence they can have on boys' and girls' development.

In recent years, children's books about girls and women who excel in STEM-related fields have become touted as a tool in combatting gender stereotypes in the popular media [17]. For example, a story book about Ada Lovelace, a pioneering female programmer, became an international bestseller according to Amazon. Presumably, part of their popularity can be ascribed to the perception that such books help to combat gender stereotypes. Recent research reveals that more frequent exposure to stereotypical content on television is correlated with more traditional gender role attitudes [18]. However, little systematic research has actually evaluated whether brief exposure to stereotype-related content in stories causally increases or decreases implicit gender stereotypes in children. In the current paper, we thus focus on the capacity to shape children's implicit *math-gender stereotype* through vivid stories that utilize exemplars who portray either stereotype-consistent, neutral, or stereotype-inconsistent roles (e.g., girls excelling and engaging with math).

## Implicit stereotypes in response to counter-stereotypic content

Evidence for the malleability of children's implicit gender stereotypes in response to brief exposure to content is sparse to date, but work in adults yields important insights. Although some theorists have argued that implicit associations are slow to change in response to one-time exposure to novel information [19], substantial work indicates that some methods are able to change implicit associations at least temporarily among adults [20]. For example, in adults, real-life contact with individuals who defy stereotypes seems to effectively weaken gender stereotypes [7]. Particularly relevant to the current study, lab-based interventions that exposed adult participants to information about counter-stereotypical exemplars (i.e., in the form of stories about such individuals) were effective at changing adults' implicit racial biases, and even more effective than other interventions [20–23]. Given the popularity of children's books as a means of communicating information to young children, it is important to empirically test whether stereotype-relevant information embedded in short stories can shape children's implicit stereotypes.

There is emerging evidence that children's implicit cognitions can change in response to newly presented information. Recent studies suggest that children can rapidly acquire implicit attitudes about novel groups in response to verbal statements (e.g., "Squarefaces are good and Longfaces are bad") [24] and the brief repeated co-occurrence of novel categories and properties [25]. Indeed, these studies suggest that children's implicit associations are formed similarly for social and non-social categories and for evaluative and stereotypic associations following a very brief exposure to vignettes depicting that content [24, 25]. While it remains unclear how long those newly formed associations persist, the data clearly demonstrate that new implicit associations can be shaped quickly from a young age. Given that this work evaluated the ability to shift implicit associations about novel groups, it is less clear whether children's existing associations for a known social category could be changed following a brief exposure to vivid stories. Existing stereotypes that are supported by past experiences and prevailing cultural norms might be less malleable to stereotype-relevant content.

Some research in children has tried to change implicit associations about existing groups. Although not in the domain of gender, and concerning attitudes rather than semantic stereotypes, research has examined stories that introduce characters who defy racial biases. In two studies, researchers found that exposure to short stories about positive Black exemplars reduced implicit white = good (and black = bad) attitudes in elementary school-aged children for up to an hour after exposure [26, 27]. While this evidence suggests that counter-stereotypic information can shape children's implicit race biases, it remains an open question whether girls' and boys' implicit math-gender stereotypes could be similarly shaped by stereotypical or counter-stereotypical exemplar stories. Particularly, gender stereotypes merit separate investigation because children's understanding of gender and race show different developmental trajectories [28]. If counter-stereotypic information embedded in a brief story can effectively shape children's implicit math-gender stereotypes, such an easily scalable intervention has potential for being applied broadly.

To our knowledge, only one previous study has examined methods of experimentally shaping implicit gender stereotypes in childhood. In a sample of 6-year-olds, Galdi and colleagues [16] changed girls' implicit math = male stereotypes through a coloring prime. Girls who colored a picture showing a girl succeeding (and a boy failing) at solving a math problem exhibited weaker math = male associations than girls who colored a control picture, or a picture implying female failure and male success. Importantly, effects of priming were only found in girls and not boys, and simply priming (stereotypically) low math performance by a girl in this particular way did not significantly increase implicit stereotypes. Whereas this evidence

suggests that exposure to counter-stereotypical information in the form of coloring book pictures can shape implicit math-gender stereotypes (at least in girls), effects of counter-stereotypical exemplars embedded in children's stories are not yet known. In addition, there is a growing emphasis on conceptual replications and high-powered studies to answer important questions in psychological science [29, 30]. Thus, further investigations with larger samples are needed to better understand whether, and to what extent, exposure to stereotypic vs. counter-stereotypic exemplars in stories can shape childrens' implicit gender stereotypes.

## Age-related changes in malleability

The second goal of the current research was to examine whether exposure to stereotype-relevant stories equally impacts the implicit gender stereotypes of children in different age groups. An important factor in understanding the flexibility of implicit gender stereotypes is to examine the developmental trajectory of sensitivity to stereotype-relevant information. There is some reason to believe the way younger vs. older children process counter-stereotypical information is qualitatively different. On the one hand, one might expect that implicit stereotypes could be more sensitive to story-type interventions in younger children as these children have learned less information in their lives which needs to be overwritten [31]. On the other hand, some might argue that younger children might be less sensitive because they have not yet developed the cognitive capabilities to quickly extract a rule like "girls = math" from hearing stories about novel male and female characters [32–34]. Gonzalez and colleagues' [26] evidence, though tentative by their own admission, suggests racial biases were only responsive to exemplar stories in children ages 9–12, and not in 5–8-year-olds. However, as gender is more salient to children than race [35], these patterns might not apply to the ability of stories to shape children's implicit gender stereotypes. As gender tends to be more easily recognized and processed by children than race is [35], it is theoretically plausible that stereotype-relevant information about gender could be internalized even by young children that do not yet readily internalize information about race. Given these mixed findings, it is important to examine whether implicit gender stereotypes would be sensitive to stereotype-relevant stories *earlier* than racial attitudes. Our second aim was thus to test whether effects of stereotype-relevant stories in children's implicit math-gender stereotypes were moderated by age-group like findings on race.

## Impacts on the self-concept

A secondary goal of the current study was to examine whether exposure to stereotype-relevant stories about math and reading would lead to corresponding changes in *math self-concept*. Implicit math self-concept is defined as the association between oneself and the domain of math [8]. If children's implicit gender stereotypes are responsive to stereotype-relevant stories, it is important to consider possible downstream consequences for their math self-concept. Balanced identity theory posits that group identity, ingroup stereotypes, and self-concept form a non-contradictory set of associations, such that attributes that we associate with ourselves should not contradict attributes we associate with groups that we incorporate into our identity [3, 8, 36–38]. Such balance is already evident by age 6, as girls at this age identify less with math to the extent that they associate math = male [3, 4]. Even though the theoretical model of balanced identity theory suggests a direct causal link between changing stereotypes and changing self-concepts, no research to date has directly examined whether *experimentally* changing implicit gender stereotypes (e.g., through exemplar exposure) directly affects children's (or even adults') math self-concepts. Since evidence suggests that the correlation between stereotypes and self-concept increases with age [3, 4], it is possible that a stereotype intervention

might not yet show an immediate and direct effect in children's self-concepts. Nonetheless, it is important for research targeting changing implicit biases to understand whether, how, and when those changes in implicit cognitions about a group lead to changes in a person's self-concept, especially given that gendered self-concepts are likely an important underlying factor in ingroup stereotypes that guide choices and preferences [36, 38]. As such, we examined whether experimentally changing the strength of math-gender stereotypes has immediate effects on children's math self-concept.

### Current research

In a sample of children aged 6 to 11, we investigated the effects of exposure to stereotype-relevant stories. Children were exposed to either two stereotypical (boys who like math and girls who like reading), two counter-stereotypical (girls who like math and boys who like reading), or two neutral (control) stories, before completing a measures of implicit math = male stereotypes, as well as measures of implicit and explicit math = me self-concept.

Our first aim was to test the effect of this manipulation on children's implicit gender stereotypes. Our central prediction was that implicit stereotypes would be responsive to these stories, such that children exposed to the *counter-stereotypical* exemplar stories would have weaker implicit math = male stereotypes than those exposed to the stereotypical and the control exemplars. Based on evidence that children are still learning gender stereotypes until early adolescence [15] coupled with findings that exposure to stereotypical exemplars did not increase racial bias among children [27], we had no clear expectation as to whether *stereotypical* exemplars would or would not increase implicit gender stereotypes above the control condition. Based on past research showing that only older but not younger children's racial stereotypes were affected by exposure to counter-stereotypical exemplars [27], we also examined whether children's age would moderate the effects of our manipulation.

A secondary aim was to test whether stereotypical and counter-stereotypical exemplars affected children's explicit and implicit math self-concept. Because past research suggests that the link between implicit stereotypes and self-concepts might be weak in young children, and strengthens throughout adolescence [3], we were curious to examine whether the impact of our brief exposure to stereotype-relevant exemplars might be sufficient to directly shape children's self-concept.

## Materials and methods

### Participants and procedure

A total of 389 children between the ages 6 and 11 were recruited at a science center in a large North-American city in the Pacific Northwest. Fifty-three children were excluded due to attention issues (13), technical issues (12), diagnosed developmental disabilities (8), researcher errors (8), non-completion (7), poor English skills (3), or having parents interrupt the manipulation (2), which resulted in a still large final sample of 336 children (169 girls, 167 boys, $M_{age}$ = 8.88, $SD$ = 1.09). Analyses degrees of freedom vary very slightly because of missing data. This final number is considered fairly large among community-samples of children [24, 26, 27]. Our sample consisted of White/European (47.6%), East Asian (27.1%), South Asian (2.7%), Middle Eastern (3.3%) and children of other ethnic backgrounds (19.3%). While parents of the children in the study did not directly report demographic information, visitors of the particular science center within which recruiting and testing took place have a median income of $75,000, with approximately 85% of parents holding at least a high school degree, and 57% holding at least a bachelor's degree. Parents were given the option to fill out a supplementary questionnaire with measures not relevant to this manuscript (see S1 File). The current work

has been approved by the behavioral research ethics board of the University of British Columbia (ID: H10-00147; Title: The development of social cognition). Data and experiment files are openly shared (https://osf.io/2h3xu/).

**Story vignettes.**   Each participant was semi-randomly assigned to one of three conditions: stereotypical, counter-stereotypical, and a neutral/baseline control (see the S1 File for full text). Participants were randomly assigned to the stereotypical vs. counter-stereotypical condition. Baseline responses were established in follow-up data-collection in the same location and by the same research assistants. Analyses suggest conditions did not significantly differ in age, $F$ (2, 314) = 0.363, $p$ = .696, gender proportion, $\chi^2$ = 0.95, $p$ = .623, and proportion of white vs. non-white participants, $\chi^2$ = 4.21, $p$ = .122. In each of three conditions, the research assistant read aloud two vignettes one at a time to the participants, instructing participants to imagine knowing the characters. For each story, there was one male (Max or Eric) and one female (Mary or Emily) character. In order to represent the largest ethnicities in our community [39], one story featured White/European appearing characters (39.2% in community) and the other featured East Asian appearing characters (36.2% in community). Each vignette described a boy and a girl engaging in and preferring an activity in childhood and then continuing to engage in that same activity through adulthood.

*Stereotypical condition.* In the stereotypical condition, the boys in both vignettes were described as preferring and performing well in math and growing up to have math-related occupations (e.g., math professor), while the girls were described as preferring and performing well in reading and growing up to have literature-related occupations (e.g., writer).

*Counter-stereotypical condition.* In the counter-stereotypical condition, the story text was identical to the stereotypical condition, only with the gender of characters reversed, meaning girls were associated with math and boys with reading.

*Baseline control condition.* In the control condition, swimming and tennis replaced math and reading (gender of characters was counterbalanced between activities) in stories of similar structure and length. These sports were chosen to avoid math or reading content, and because they are relatively popular among men and women [40]. Pilot data from a separate sample indicated that both sports were perceived to be similarly liked by both genders (see SI).

## Measures

**Manipulation checks.**   After each vignette, the participants were asked four questions designed to assess their comprehension of the vignettes, of which children answered on average 98.17% correctly (details in SI).

**Implicit association measures.**   Two child-friendly Implicit Association Tests (Child IATs) [41] were used to assess children's implicit math-gender stereotypes and their implicit math self-concept respectively. The order of these two tasks was fixed (stereotypes then self-concept) because we were primarily interested in the effect on implicit stereotypes. The IAT, a well-validated measure with adults [42] and children [14, 31, 37, 43], assesses the strength of associations between concepts by measuring the speed and accuracy with which participants categorize stimuli into stereotype-congruent vs. stereotype-incongruent category pairings.

*Implicit math-gender stereotype IAT.* For the math-gender stereotype IAT (see Fig 1 for a schematic of the procedure), participants were first familiarized with the stimuli from the target (boys vs. girls) and attribute (math vs. reading) categories separately in 12 trials each. So, for 12 trials, participants were instructed to sort stimuli (images of individual boys and girls) into the categories "boy" and "girl" using two response buttons. For another 12 trials, participants were instructed to sort stimuli (words related to math and reading) into the categories

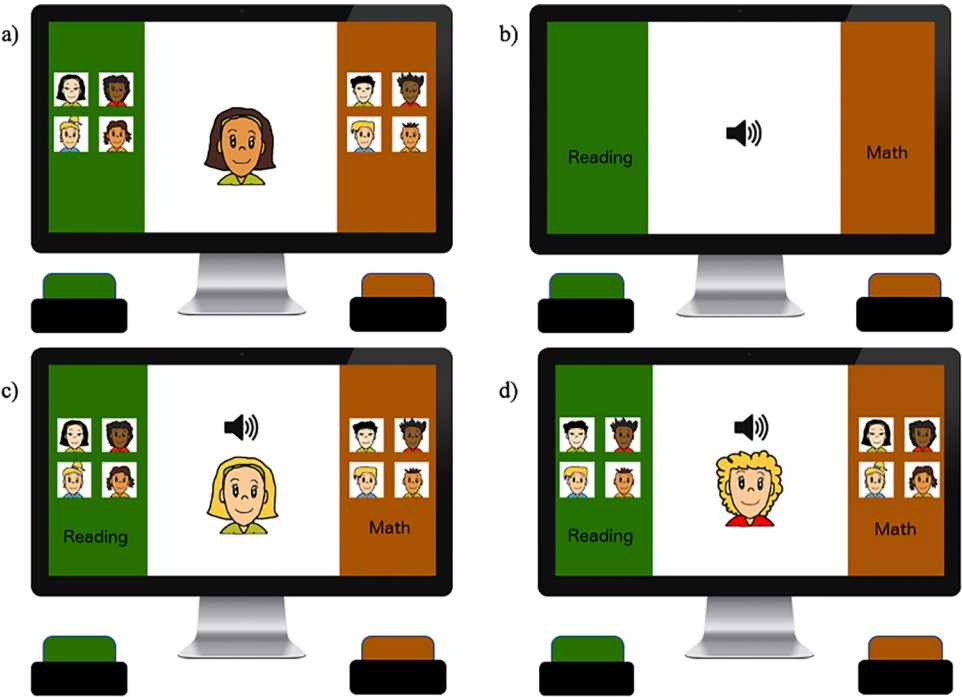

**Fig 1. Schematic of implicit stereotype measure.** A) and b) practice blocks: participants categorize only stimuli from the target categories (gender) or the attribute categories (math vs. reading) separately. C) and d) critical blocks: participants categorize stimuli from both target and attribute categories, one at a time, with two categories each mapped on one of two response buttons in stereotype-congruent (c) vs. stereotype-incongruent pairings (d).

"math" and "reading" using two response buttons. On each trial, a stimulus was presented and children made categorizations of the stimulus into the target or attribute categories displayed on the left and right side of the screen using two colored JellyBean™ response buttons. Each response button had a unique color (either red or green) which corresponded with the side of the screen where the category labels were placed (either red or green).

Stimuli for the gender categories were cartoon images of boys and girls, which varied in skin tone, eye, and hair color to represent an ethnically diverse sample (for each gender we had White/European, East Asian, Black, and Latinx-appearing exemplars). The stimuli for the attribute categories consisted of math-related (addition, count, math, numbers) and reading-related (books, letters, words, read) words that were presented acoustically [14]. As with past studies with children using this methodology, a red "x" appeared whenever a stimulus was categorized incorrectly and disappeared once participants made the correct response in the practice blocks, but no "x" was presented in the critical blocks (children making a mistake were simply prompted by the researcher to try again).

Upon completing the two practice blocks, the participants began the first of two critical test blocks ($N = 40$ trials), requiring them to categorize stimuli from both target (gender) and attribute (math vs. reading) categories using the same two response buttons. In critical blocks, one target group and one attribute group were mapped onto a single response button whereas the other target group and attribute were mapped onto the second response button. In critical blocks, targets and attributes were paired either in a stereotype-congruent (math+male vs. reading+female) or stereotype-incongruent (math+female vs. reading+male) manner. After the first critical block, participants completed another practice block, in which the math and reading categories switched sides on the screen ($N = 20$ trials). Afterwards, in the second

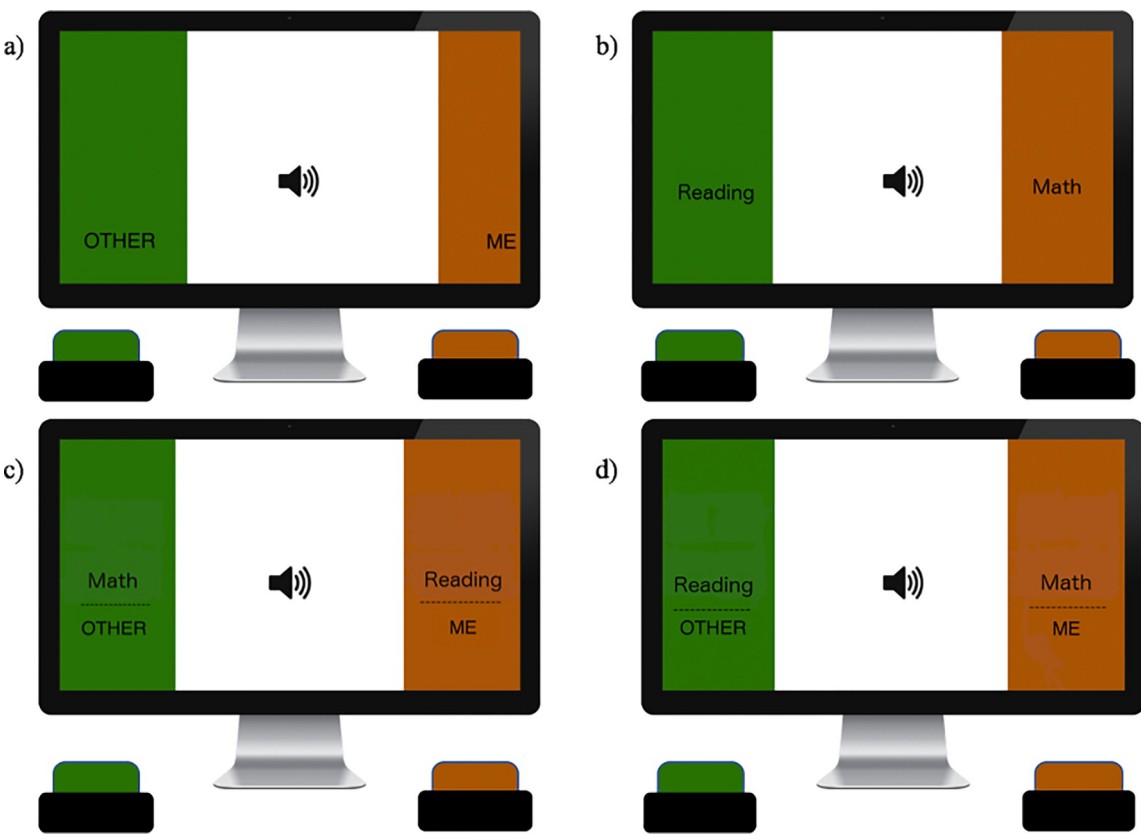

**Fig 2. Schematic of implicit math self-concept measure.**

critical test block ($N$ = 40 trials), the participants categorized both target and attribute stimuli again, with pairing of target and attribute always being opposite to what the child had seen in the first critical block (order of critical blocks counter-balanced). Higher scores on this measure indicated that children were faster to respond when math was paired with male rather than female stimuli, indicating a stronger association between math = boys (and reading = girls).

*Implicit math self-concept IAT.* A second Child IAT was used to assess the extent to which children themselves implicitly identified with math vs. reading (see Fig 2). For this IAT, participants were asked to sort math vs. reading stimuli identical to the stereotype measure in addition to stimuli from the target categories "me" (I, mine, my, myself) and "other" (their, them, themselves, they) [14, 37]. Similar to the math and reading words, these stimuli were also presented acoustically. Higher scores on this measure indicated that children were faster to respond when math, rather than reading, was paired with "me", thus indicating an association between math = self (and reading = others).

**Explicit math self-concept.** To assess children's self-reported identification with math, we included two questions assessing liking of math ("How much do you like to do math?"; "How fun do you think it is to do math?"), and two questions assessing math self-efficacy ("How good are you at learning new things about math?"; "How good at math are you?") after the implicit measures. Participants were asked to indicate their answers by pointing to one of five circles placed in ascending size from left to right with the response labels (e.g., "not at all" to "a lot"). Because of high correlations, responses to these items were averaged to provide an index of explicit math self-concept ($\alpha$ = .78).

## Results

### Data preparation

For each IAT we calculated D-scores following the procedures outlined by Greenwald and colleagues [44] and Baron et al. [40] and used extensively in developmental studies of implicit bias [31, 36, 45]. According to recent recommendations for reaction time measures of implicit associations [46], we excluded children who had more than 10% of trials below 300ms ($n_{\text{stereotypeIAT}} = 0$; $n_{\text{self-conceptIAT}} = 2$) and children who had made errors on more than 30% of trials ($n_{\text{stereotypeIAT}} = 18$; $n_{\text{self-conceptIAT}} = 25$) on analyses in which the given measure was used (i.e., children who failed self-concept IAT criteria were not excluded from analyses on the stereotype IAT if they passed these criteria). Means, standard deviations, and correlations for all key variables by child gender are shown in Table 1. After all exclusions, we excluded 18.35% of the original sample (N = 389) and ended with a final sample of 318 for our key analyses on implicit gender stereotypes. This exclusion rate is typical of studies with children using the IAT conducted in museum settings (Gonzalez et al., 2016, 2017, 2021).

### Effects of vignette exposure on stereotypes

To examine whether our story vignettes significantly affected children's math-gender stereotypes, we conducted a 2 (gender: boy vs. girls) x 3 (condition: stereotypical, control, counter-stereotypical) ANCOVA with implicit math = male stereotypes as the dependent variable. Analyses controlled for child age ($z$-scored) as a covariate. As hypothesized, we found a significant main effect of condition, $F(2, 311) = 6.28$, $p = .002$, $\eta^2 = .04$, suggesting that brief exposure to stereotype-relevant story vignettes are able to effectively change children's implicit gender stereotypes about math.

As displayed in Fig 3, follow-up simple comparisons revealed that participants exposed to counter-stereotypical story vignettes showed significantly lower implicit stereotypes than those exposed to stereotypical story vignettes, $SE_d = .05$, $p < .001$, $d = -.49$. Though implicit math-gender stereotypes were not significantly weaker in the counter-stereotypical compared to the baseline control condition, this difference was not statistically significant, $SE_d = .05$, $p = .126$, $d = -.21$. Importantly, implicit math-gender stereotypes were significantly *heightened* in children exposed to the stereotypical vignettes compared to the control story vignettes, $SE_d = .05$, $p = .036$, $d = .30$, suggesting that stereotypical information increased implicit stereotypes over and above levels in the control condition. Counter to a previous finding by Galdi and colleagues [16], condition did not interact with gender, $F(2, 311) = 1.42$, $p = .243$, $\eta^2 = .009$, suggesting that our story vignettes shaped implicit gender stereotypes in similar ways for boys and girls.

**Table 1. Correlations and descriptive statistics for key variables ($N_{\text{girls}} = 151$; $N_{\text{boys}} = 145$).**

| Variables | 1 | 2 | 3 | 4 |
|---|---|---|---|---|
| 1. Implicit Math = Male Stereotype | - | .07 | -.10 | .08 |
| 2. Implicit Math Self-Concept | -.15† | - | -.03 | -.07 |
| 3. Explicit Math = Male Stereotype | -.05 | .08 | - | -.11 |
| 4. Age | .04 | -.07 | -.06 | - |
| Mean_girls (SD) | .16 (.35) | .09 (.38) | 3.50 (0.93) | 9.98 (1.05) |
| Mean_boys (SD) | .07 (.37) | .13 (.32) | 3.73 (.92) | 8.95 (1.09) |

*Note*. Correlation values above diagonal represent values for boys, while values below diagonal represent values for girls

† $0.05 < p < 0.10$.

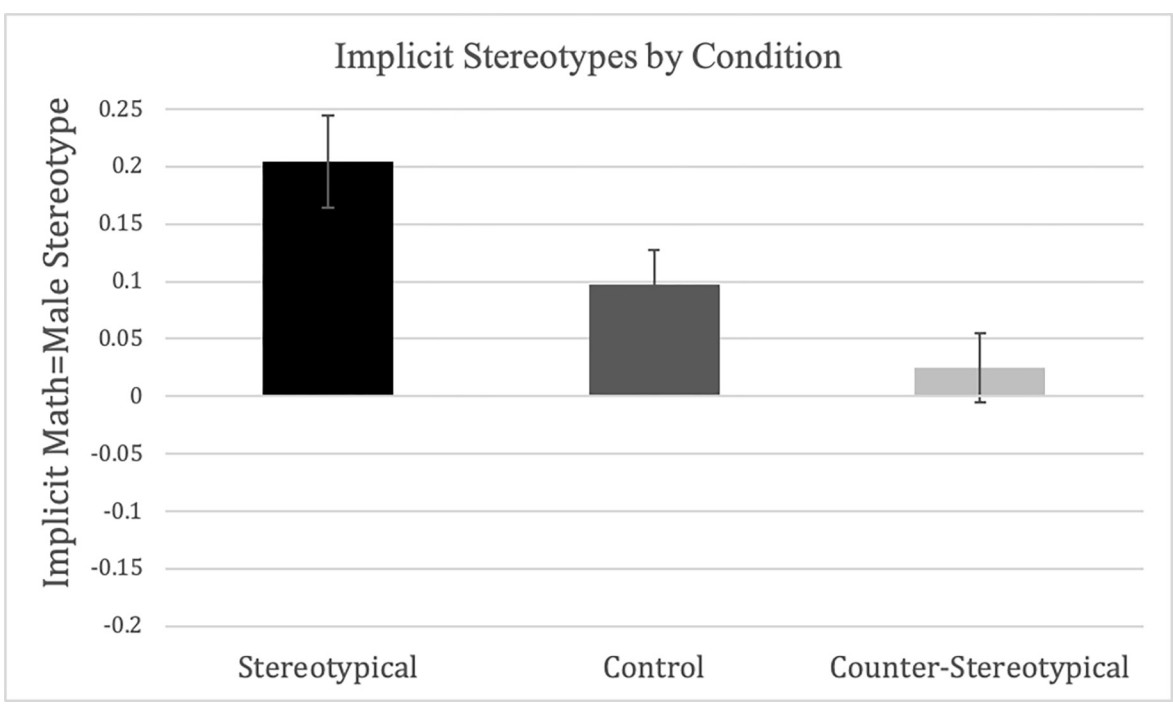

**Fig 3. Implicit stereotypes by condition.** Error bars represent standard error of the mean. Scores above zero denote an association of math = male (& reading = female), scores below zero denote the opposite pattern.

Further follow-up one-sample $t$-tests against zero (which indicates no clear association), tested whether participants showed evidence of significant gender stereotyping within each condition. These analyses were collapsed across gender and age as neither term interacted with condition. Results suggested that, as expected, and similar to children in previous research [14], children in the control condition, $M = .10$, $SD = .34$, $t(114) = 3.06$, $p = .003$, significantly associated math more with boys than with girls. Furthermore, children in the stereotypical condition, $M = .20$, $SD = .39$, $t(99) = 5.23$, $p < .001$, exhibited an even stronger and significant association between math and boys (and between reading and girls). In contrast to both of these conditions, children who were exposed to counter-stereotypical story vignettes were the only group who showed no significant implicit math-gender stereotypes, associating neither gender more strongly with math, $M = .02$, $SD = .34$, $t(102) = 0.74$, $p = .460$. Taken together, these results support our hypothesis that implicit gender stereotypes are malleable in children in response to a brief story-based intervention.

Finally, a main effect of gender was observed (but no condition x gender interaction). This analysis revealed that, across conditions, girls had stronger implicit math = male stereotypes than did boys, $F(1, 311) = 4.30$, $p = .039$, $\eta^2 = .01$. Yet, one-sample $t$-tests against zero within each gender (and collapsing across conditions) indicated that both girls, $t(161) = 5.50$, $p < .001$, and boys, $t(155) = 2.05$, $p = .042$, showed significant levels of implicit math-gender stereotypes (i.e., significantly associating math with male more than with female)

### Exploratory effects by age group

While we had no *a priori* predictions about effects of age, we conducted exploratory analyses to examine whether the above effects differed markedly between younger ($M_{age} = 7.96$, $SD = 0.57$, $n = 167$) and older children ($M_{age} = 9.79$, $SD = .60$, $n = 169$). Given that past

research suggested qualitative differences in the effects of counter-stereotypical exemplars in younger vs. older children, these groups were defined by a median split on age in our sample at 8.80 years, which was similar to the median split age in Gonzalez and colleagues' (Split at 8.39) analyses on racial attitudes [26]. This split was also congruent with age-groupings on previous work on children's implicit intergroup biases [25, 27].

To examine whether our story vignettes affected children's math-gender stereotypes differently across age groups, we conducted a 2 (gender: boy vs. girls) x 3 (condition: stereotypical, control, or counter-stereotypical) x 2 (age group: younger vs. older) analysis of variance with implicit math = male stereotypes as the dependent variable. There was a significant effect of age group $F(1, 306) = 5.66$, $p = .018$, partial $\eta^2 = .02$. Consistent with past evidence [3], older children ($M = .15$, $SD = 0.36$) had stronger implicit stereotypes than did younger children ($M = .06$, $SD = 0.36$), with older children's stereotypes significantly greater than zero, $t(164) = 5.51$, $p < .001$, but younger children's only marginally so, $t(152) = 1.97$, $p = .051$. However, results revealed no significant two- or three-way interactions between condition, age, and gender, partial $\eta^2 < .01$, $Fs < 1.36$, $ps > .256$. The previous main effects of condition and gender remain significant when entering age group as the additional variable in the analyses. Additional exploratory analyses with math self-concept as outcome can be found in the SI.

Even though we found no formal interactive effects of age and condition (or gender) on implicit gender stereotypes, we saw it as important to examine older and younger kids' implicit stereotypes in response to our story vignettes separately, given mixed previous research suggesting that implicit racial stereotypes can be changed in older but not younger children using a similar experimental manipulation [26, 27]. As displayed in Fig 4, separate analyses of the main effects of condition on implicit gender stereotypes (controlling for gender as in other analyses) suggested clearer evidence for an effect in younger children, in contrast to Gonzalez and colleagues' findings [27]. Specifically, these analyses yielded a significant main effect of condition on our younger sample, $F(2, 147) = 7.08$, $p = .001$, $\eta^2 = .09$, but not our older sample, $F(2,159) = 1.08$, $p = .341$, $\eta^2 = .01$. Simple comparisons in younger children showed that, as in the main analyses, children in the stereotypical condition had stronger stereotypes than did children in the counter-stereotypical, $SE_d = .07$, $p < .001$, and the control condition, $SE_d = .07$, $p = .023$. While in the same direction, no simple comparisons were significant in the older group, $ps > .16$. Additional analyses with age as continuous variable yield largely similar patterns and are detailed in the SI. Specifically, while there no was no formally significant interaction between continuous age and condition (p = .092), effects of condition on implicit stereotypes were significant for younger children in our sample (-1SD on age), but non-significant in older children in our sample (+ 1SD). Detailed analyses can be found in the SI.

### Effects of vignette exposure on self-concept

**Implicit self-concept.** To examine whether our story vignettes significantly affected children's math self-concept, we conducted a 2 (gender: boy vs. girls) x 3 (condition: stereotypical, control, counter-stereotypical) ANCOVA with implicit math self-concept as the dependent variable. Analyses controlled for child age (*z*-scored) as a covariate. See Table 1 for means. Age did not significantly moderate the effect of condition on self-concept. See S1 File for details.

As displayed in Fig 5, despite patterns going in the hypothesized direction for girls, there was no main effect of condition, $F(2, 302) = 1.02$, $p = .363$, $\eta^2 = .007$, or condition by gender interaction, $F(2, 302) = 1.05$, $p = .351$, $\eta^2 = .007$, on implicit self-concept. Main effects of gender and age were also non-significant, $Fs < 1.18$, $ps > .27$. Across conditions, single-sample *t*-tests against zero showed that both girls, $t(156) = 3.11$, $p = .002$, and boys, $t(151) = 4.88$, $p < .001$, significantly identified themselves with math more than with reading (see Table 1).

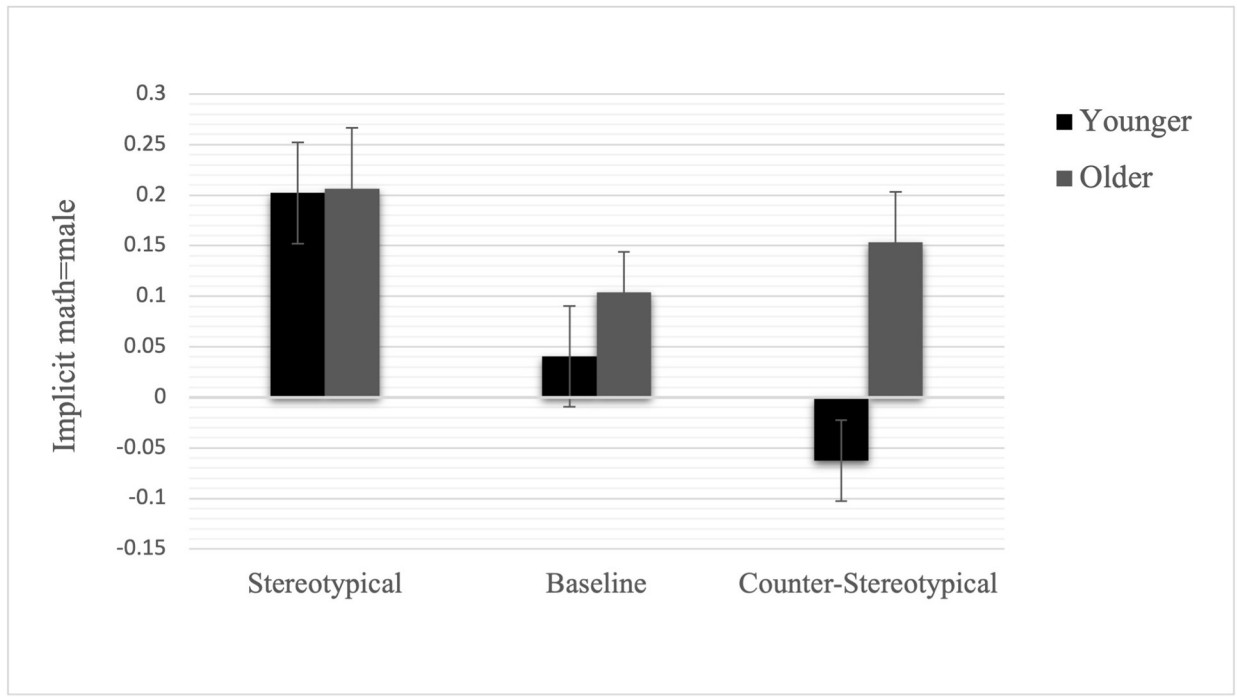

**Fig 4. Implicit stereotypes by age-group and condition.** Error bars represent standard error of the mean. Scores above zero denote an association of math = male (& reading = female), scores below zero denote the opposite pattern.

**Explicit self-concept.**　Boys reported non-significantly higher explicit math self-concept than did girls, $F(1, 329) = 3.29$, $p = .071$, $\eta^2 = .01$. Yet, no main or interactive effects were observed for condition or age, $F < 0.59$, $ps > .45$. Similar to findings on the implicit math self-concept, single sample $t$-tests against the midpoint of the scale (3) suggested that both girls, $t$

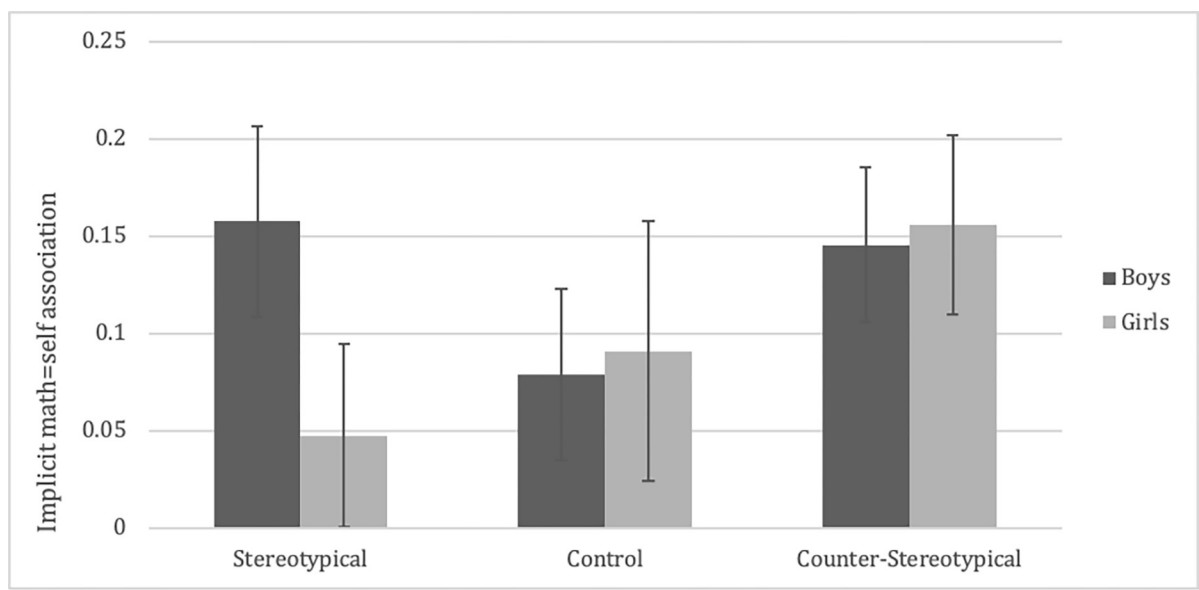

**Fig 5. Implicit self-concept by condition and gender.** Error bars represent standard error.

(168) = 7.24, *p* < .001, and boys, *t*(166) = 9.93, *p* < .001, in our sample explicitly identified with math over reading (see Table 1).

## Discussion

In line with previous work [3, 4], findings from this large sample of children suggested that by age six, boys and girls show evidence of potentially harmful implicit stereotypes that associate math with boys more than with girls. Given the myriad of negative downstream consequences of such gender stereotypes, it is important to identify methods that can change such associations in childhood, before they markedly shape education trajectories. Despite the recent popular media articles promoting the power of children's books to change stereotypes, there is little previous research to document whether brief exposure to stories with counter-stereotypical vs. stereotypic content would actually change children's implicit gender stereotypes. In a well-powered study, we found clear evidence that brief exposure to stories with a stereotypical vs. counter-stereotypical protagonist does indeed affect children's implicit math-gender stereotypes. Children who heard stories about math-oriented girls and reading-oriented boys were the only group in this experiment not to show a significant implicit math = male stereotype, whereas those in a control condition, and those assigned to a condition reinforcing cultural stereotypes, did show this bias. Consistent with the work on implicit race attitudes in children [26, 27] and adults [47], we observed that associations did not significantly *reverse* after exposure to counter-stereotypical exemplars. Such patterns are to be expected since one-time exposure to counter-stereotypical exemplars has to compete with a wealth of prior information facilitating stereotypical associations between math and girls.

Our finding that implicit math = male stereotypes significantly *increased* after exposure to stereotypical exemplars compared to our control condition, underscores that implicit math-gender stereotypes are malleable in childhood, and potentially shaped by stereotype-relevant information. This insight critically underscores that childhood likely constitutes an important time in which gender stereotypes form increasing readily in response to stereotype-relevant information. Coupled with initial evidence that TV exposure correlates with gender role beliefs in children [17], our experimental findings might suggest that limiting children's exposure to stereotype-affirming content might be just as (or more) important that exposing them to counter-stereotypical information. This pattern is especially noteworthy because attempts to change racial biases in adults and children did *not* result in an increase in existing pro-white bias [22, 27].

One limitation of our design is that data in the neutral condition was collected a few months after the data of the stereotypical and counter-stereotypical condition. While we took great care to collect the neutral condition with an age- and gender-matched sample, in the same setting, and with the same research assistants, this non-random assignment to the control condition raises a small possibility of non-experimental effects. While differences between the concurrently collected conditions are significant and demonstrate effects of exposure to exemplars, differences from baseline could be somewhat more tentative. A replication study could demonstrate differences from baseline with greater certainty. Nonetheless, these data point to the possibility that it may be critical to consider developing strategies that focus on reducing exposure to stereotypic information (and not just on increasing exposure to counter-stereotypic information) in childhood.

Results of this study also showed intriguing age-related effects that call into question whether children's readiness to process information about race and gender have the same developmental trajectory. Previous work with a similar sample of children showed that implicit racial attitudes could *not* be easily changed in children ages 5–8, but were malleable a bit later

in development [26, 27]. In contrast, we find that children's implicit math-gender stereotypes were affected by exposure to stereotype-relevant stories during this younger age range. While we did not find a significant condition by age interaction, exploratory analyses showed that the effect of the story vignettes on children's implicit stereotypes was significant only when looking at the younger children in our sample (6 to 8.8 years). In fact, effects were descriptively stronger among younger children than among the older children in our sample. Future research must replicate these exploratory findings. At the least, however, these findings suggests that implicit gender stereotypes are not only responsive to stereotype-relevant content in a story at a young age, but when using counter-stereotypical exemplar exposure, they might be especially malleable in younger children compared to older children.

Future research should more closely examine the age trajectory of sensitivity to counter-stereotypic exemplars. If our results hold, there is still a question whether age-related shifts in how counter-stereotypical exemplars are processed are gradual (e.g., children might gradually become less sensitive to interventions with time) or qualitative (e.g., once children reach a certain developmental milestone, they process interventions differently). Our analyses show the same patterns when examining different age groups vs. age as continuous moderator of condition. Thus, future research is needed to determine the exact nature of the development of sensitivity to exposure to counter-stereotypical exemplars.

Relatedly, future work must also determine *why* gender stereotypes in younger children may be more responsive to exemplar information embedded in a story than racial attitudes seem to be. There are two possibilities for an earlier and stronger effect of stories on implicit gender stereotypes having to do with: (a) the ease with which children encode exemplars as representative of the category, and (b) the extent to which they view exemplars in the story as self-relevant. Consistent with the first possibility, past work shows that children pay more attention to information about gender than about race [35]. Giving some credence to the second possibility, gender is seen as more self-relevant than race by some adolescents [48] and children [49], and we learn self-relevant novel stimuli faster than non-relevant counterparts [50]. Given the salience and self-relevance of gender, younger children in our study might have more effectively encoded information from our exemplar intervention, compared to similar interventions designed to change race attitudes. Indeed, with an eye towards improving the longevity of changes in implicit cognitions, future work will need to investigate exactly what "ingredients" make exemplars most effective.

Future work will also need to determine how long the changes in implicit gender stereotyping as a result of this brief intervention persist. Work on the effectiveness of online interventions on changing adult's implicit race bias suggest that, though initially effective [22], effects of numerous interventions dissipate completely after 24 hours [23]. In contrast, recent evidence suggests that story-based counter-stereotypical exemplars still measurably reduced children's implicit race biases an hour after the intervention [27]. Given how uncertain the durability of stereotype-change is in children, future work may want to investigate how long-lasting effects of a single brief manipulation like ours is on children's implicit math-gender stereotypes.

However, given the repeated nature of exposure to stereotypical content in the real world, statistically significant changes in response to our brief content could potentially be very meaningful. With longer and/or repeated exposure to stories that counter gender stereotypes, effects could become stronger and/or more lasting. For example, recent research suggests that exposure to counter-stereotypical content in television programs reduces children's racial biases for up to four weeks later [51]. If children own several counter-stereotypical story books, like the popular book about Ada Lovelace cited in the introduction, and read them repeatedly, it is possible that their effects could strengthen and solidify with repeated exposure.

A more exploratory goal of the current work was to assess the causal effects of stereotype change on children's math self-concept. In light of past work showing that the link between implicit ingroup stereotypes and self-concept strengthens in adolescence [3], our non-significant effects of exemplars on self-concept could be due to our participants lacking full cognitive balance between ingroup stereotypes and self-concept at the age examined. Given the dearth of research on how children's self-concepts are causally shaped by implicit stereotypes at an early age, future research should focus specifically on the effect of stereotype-changing interventions on children's self-concepts. For example, it is possible that self-relevance of stories needs to be increased to affect children's self-concept.

It is worth noting that, overall, we did not find significant gender differences in either explicit or implicit math self-concept. In addition, we found that effects of condition on implicit and explicit math self-concepts were not different for boys vs. girls (see S1 File for details). Whereas gender differences in math self-concept persist in adolescents and adults [52], recent studies have found mixed results on gender differences in math performance and self-concept, with some studies showing no gender difference or reversed gender differences (i.e., greater math self-concept in girls) in young children [e.g., 53, 54], especially in some cultural contexts [55]. This could either be a sign of a positive societal trend or simply mean that gender stereotypes about math are not internalized and applied to the self-concept until a later age. Our sample in particular could have somewhat obscured gender differences in math self-concept at baseline because all children were tested in a science museum that presumably disproportionately attracts parents and/or children with an interest in STEM fields. To better estimate true gender differences in math self-concept in children, a large fully representative sample would be more appropriate.

Implicit gender biases play a key role in gender inequalities in highly valued STEM fields [3, 4, 6, 8, 9, 11]. In a large community sample, we provide evidence that young children's implicit math-gender stereotypes are changed by a brief exposure to stereotype-relevant content. Through exposure to brief vivid stories, we provide evidence that implicit math gender stereotypes can be both eliminated and increased early in development. Our findings contribute to a theoretical understanding of the malleability of implicit bias across development, but also provide practical guidance for how cautious we must be with sensitive messages to young children in the domains of gender and math. In practice, our findings suggest that if we have the goal to reduce harmful math-gender stereotypes in children, increasing exposure to others who defy such stereotypes, even through simple stories that children hear, appears a promising avenue. Critically, however, our findings especially demonstrate stereotype-affirming content can measurably exacerbate children's existing gender stereotypes. In practice, this would suggest that limiting exposure to content that clearly perpetuates stereotypes (e.g., children's books with stereotypical protagonists) might be extremely important, as seeing such exemplars strengthened children's stereotypes in our study.

## Supporting information

**S1 File. Supporting information.** Additional detail on methods and supplementary analyses. (DOCX)

## Author Contributions

**Conceptualization:** Katharina Block, Antonya Marie Gonzalez, Clement J. X. Choi, Zoey C. Wong, Toni Schmader, Andrew Scott Baron.

**Data curation:** Katharina Block, Clement J. X. Choi, Zoey C. Wong.

**Formal analysis:** Katharina Block, Antonya Marie Gonzalez.

**Funding acquisition:** Toni Schmader, Andrew Scott Baron.

**Investigation:** Katharina Block, Clement J. X. Choi, Zoey C. Wong, Andrew Scott Baron.

**Methodology:** Katharina Block, Antonya Marie Gonzalez, Andrew Scott Baron.

**Project administration:** Katharina Block, Clement J. X. Choi, Andrew Scott Baron.

**Resources:** Clement J. X. Choi, Zoey C. Wong, Andrew Scott Baron.

**Software:** Clement J. X. Choi, Andrew Scott Baron.

**Supervision:** Katharina Block, Andrew Scott Baron.

**Validation:** Katharina Block.

**Visualization:** Katharina Block.

**Writing – original draft:** Katharina Block.

**Writing – review & editing:** Katharina Block, Antonya Marie Gonzalez, Clement J. X. Choi, Zoey C. Wong, Toni Schmader, Andrew Scott Baron.

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
