## [Decision Letter · Decision Letter 0]

17 Sep 2021

PONE-D-21-23166Exposure to Stereotype-Relevant Stories Shapes Children’s Implicit Gender StereotypesPLOS ONE

Dear Dr. Block,

Thank you for submitting your manuscript to PLOS ONE. After careful consideration, we feel that it has merit but does not fully meet PLOS ONE’s publication criteria as it currently stands. Therefore, we invite you to submit a revised version of the manuscript that addresses the points raised during the review process.

The reviewers have provided very detailed feedback. Please address their concerns in your revision. 

We look forward to receiving your revised manuscript.

Kind regards,

Natalie J. Shook

Academic Editor

PLOS ONE

Journal Requirements:

2. Peer review at PLOS ONE is not double-blinded (https://journals.plos.org/plosone/s/editorial-and-peer-review-process). For this reason, authors should include in the revised manuscript all the information removed for blind review.

“YES - funders played no role in research design, analyses, or writing”

“NA”

7. Please include a separate caption for each figure in your manuscript.

Reviewers' comments:

Reviewer's Responses to Questions

**Comments to the Author**

1. Is the manuscript technically sound, and do the data support the conclusions?

Reviewer #1: Yes

Reviewer #2: Yes

2. Has the statistical analysis been performed appropriately and rigorously? 

Reviewer #1: Yes

Reviewer #2: Yes

3. Have the authors made all data underlying the findings in their manuscript fully available?

Reviewer #1: Yes

Reviewer #2: Yes

4. Is the manuscript presented in an intelligible fashion and written in standard English?

Reviewer #1: Yes

Reviewer #2: Yes

5. Review Comments to the Author

Reviewer #1: This manuscript described an intervention aimed to change children’s implicit associations of math/reading with girls/boys and their math self-concepts, or the extent to which children associated themselves with math (implicitly and explicitly). The authors find that although the gender stereotype relevant storybooks demonstrate differences by condition for the gender/math IAT outcome, there were no differences by condition for implicit or explicit math self-concept. The authors additionally explored age a potential indicator of when children are most susceptible to stereotype-relevant intervention. Overall, I found the manuscript to have strong theoretical justification and report valuable findings. However, I would encourage the authors to consider further explaining a couple methodological and analytical decisions and reconsider a few implications in the discussion.

Semi-random assignment of condition: The authors note that “Each participant was semi-randomly assigned to one of three conditions…” Presumably due to the fact that the baseline data was collected at a different time, the authors describe the design as semi-random. I would recommend elaborating on the implications of the lack of full random assignment as a limitation in the discussion.

Self-concept and math: The authors report no differences by condition for self-concept (implicit or explicit) with math. The results do indicate that both girls and boys implicitly and explicitly associated themselves with math at similar rates. It may be helpful to address the lack of gender difference in math identity within the discussion, especially given that this finding contrasts with the findings on gender/math stereotypes.

Median split for age: I was concerned by the analytical decision to perform a median split on age because median splits typically lose statistical power and can lead to spurious effects (MacCallum, Zhang, Preacher, & Rucker, 2002). It is not clear whether this decision was made to be consistent with the age groups in Gonzalez et al. (2021) or for some other theoretical reason. I would recommend incorporating additional analyses including age difference as a continuous variable (perhaps in the SOM if the median split is theoretically justified in the main text). This would allow readers a fuller picture of whether there are age-related changes in malleability.

Race and gender stereotype learning: In the discussion the authors state “Consistent with the second possibility, gender is seen as more self-relevant than race by some adolescents [47]…” Because the authors have previously argued that age is likely important for acquisition of stereotype relevant information about social groups, I would question whether citing this study of multiracial adolescents would support their claim. If the authors choose to keep this citation, it would be valuable to specify the ages of the participants and note the limits of generalizing between the cited study and the current study’s sample populations.

Minor comments:

1. On line 463 (p. 20) and line 513 (p. 22) and the same sentence is repeated: “However, given the repeated nature..”

2. Figure 4 would be stronger with error bars included.

3. There is a small redundancy on line 117 (p. 5) “…exposure to at least frequent exposure…”

4. “However, little systematic research has actually evaluated whether brief exposure to stereotype-related content in stories causally increases or decreases implicit gender stereotypes.” (line 118, p. 5) This should be qualified to be “...increased or decreases implicit gender stereotypes in children.” There is plenty of research on similar topics in adults (e.g., Dasgupta & Asgari, 2004).

Reviewer #2: Thank you for the opportunity to review research on a topic that continues to be of practical and research interest: gender and STEM, in particular, young children’s implicit gender-math stereotypes. I would summarize the paper as follows: a community sample of children (n = 336, 50% female, 50% male, mean age 9) were exposed to two stories with counterstereotypic (a girl who liked math, a boy who liked reading), stereotypic (vice versa), or control (a girl who liked tennis, a boy who liked swimming) information, and then completed two IATs (girl/boy and math/reading; self/other and math/reading) as well as a measure of explicit math self-concept. Results indicated that the counterstereotypic group had weaker male-math implicit stereotypes than the stereotypic group, and was the only group whose stereotype did not significantly differ from zero. Follow-up analyses suggested that this was true only in children younger than 8.8 years. No effects were found for explicit self-concept.

I thought that the nature of the sample (large in size, young children, community sample) was a strength of this paper, and that the measures were appropriate for this sample. Below I make recommendations for improvements.

Major issues

Introduction

1. I realized later in the paper htat the self-concept question was meant to be somewhat exploratory, but it fits awkwardly into the introduction section. For instance, on p. 3, it is presented as a third research question alongside what later appear to be the chief research questions (effects of counteresterotypic stories; moderation by age); this does not suggest that it is a lower-tier or more exploratory question. On the other hand, when I got to that part on page 3, I was surprised; I felt like an interest in “self-concept” came out of nowhere. I would recommend setting up why self-concept would be a reasonable thing to measure, even briefly, on that first page. Then, if the authors wish to “demote” that research question relative to the first two, that could be made clear in the listing of the questions.

2. On p. 4, paragraph 2, the literature review focuses heavily on examples of research with young girls. Perhaps this paragraph should not set up the idea of “across the lifespan”, unless the authors are making the case that the immediate relationships with identification & achievement have downstream consequences later in life. Otherwise, I would extend the literature covered in this section. E.g., a research example of implicit stereotypes predicting grades in college students: Ramsey & Sekaquaptewa (2011)

3. One of the central justifications for the work is never supported empirically: p. 4, line 114 “children’s books about girls and women who excel in STEM-related fields have become increasingly popular.” I’d like to see some examples (Rosie Revere, Engineer? Doc MacStuffins?), and I’d like some quantification of “increasingly popular.”

4. On p. 6, I’m not convinced that the first paragraph makes sense where it is. Work on novel stereotype acquisition does not seem to speak to what this paper is going to test, since it’s established that children of the age group you will be testing have already acquired these math-gender stereotypes. I think this work should be presented in a different framing and a different spot. In contrast, the support from citations 25 and 26 in the next paragraph is much more directly compelling evidence.

5. Nevertheless, as I continued through that paragraph on p. 6, I thought that it would make sense to add research comparing the ease of acquiring gender vs. race-based stereotypes, as my understanding was that gender-based ones might be more robust, and thus more resistant. This was addressed on p. 8, but the only citation is a review (I believe Ruble, Martin, & Berenbaum, 2006), and I think the argument would be strengthened with more primary sources, especially since the comparison between race and gender stereotypes becomes a heavy focus in the discussion later on.

6. Related to point 1,on p. 9, a lot is made of the fact that no one has experimentally tested the impact of changing stereotypes on math self-concepts. I thought, “what a huge gap, this should be emphasized more!”, but then this question is treated like an afterthought in the analysis and discussion. I recommend more coherence: perhaps this section could be tempered so that the vibe is less “we are the first to tackle this important question” and more “because this is understudied, it is not clear if we’d see those effects, maybe yes (CITE) and maybe no (CITE), and so we will examine this as well.”

Method

7. P. 11, line 254: clarify - parents reported education and income to the authors? If so, please use past tense. If this is general science center visitor data, it should be cited.

Results

8. P. 15, is citation 44 a source of recommendations for children in particular, or in general? Could add some context for how these error rates/rejection rates compare to adults.

9. My biggest question in the results is why a median split was employed. I understand that Gonzalez et al. (2020) split children into age groups, but from my reading, they did not do a median split. I’d be more convinced by some kind of developmental reason for why 8.8 is a meaningful cutoff. Otherwise, what about a regression analysis, looking at dummy coded condition interacting with age? Especially since gender didn’t matter in prior analysis, wouldn’t necessarily need to worry about three way interactions with gender. A stronger rationale for the median split is needed.

10. Another major concern I have with the paper is frequent mentions of “descriptive differences” – I don’t think these are necessary and they may imply significance where there is none. The authors might refer people to figures without adding this language.

11. I’m fascinated that girls showed implicit math self-identification, and I wonder if this is typical for girls?

Discussion

12. Several interesting arguments were made in the discussion (e.g., the need to reduce stereotypic exemplars, not just increase counterstereotypic; the two reasons offered for why gender and race stereotypes may operate differently).

13. On p. 22, I’d offer another possibility: the categories “math and reading” are more discrete than “good and bad,” perhaps also allowing for easier absorption of examples? This would be a distinction between stereotyping and prejudice more broadly, and might be worth looking into the literature for any developmental lit for that?

14. On p. 22, line 509, confusing sentence - the prior sentence seems like it is about persistence already; is there a missing word?

15. The discussion mostly focuses on the effects of single brief interventions, which makes sense because that is what was tested. But given the emphasis on the rise of counterstereotypic books, I’d like to see some discussion of what might happen if, say, a classroom read counterstereotypic books all year - can the authors draw from any field intervention work and make a prediction?

Minor issues

1. P. 4, line 98: Word choice: in your description of implicit stereotypes, “domain” doesn’t sound like an attribute to me. To me, attribute would = trait (e.g., “good at math”; something that describes the group in question); otherwise it’s just another concept I think.

2. P. 7, line 172: typo: “known”

6. PLOS authors have the option to publish the peer review history of their article (what does this mean?). If published, this will include your full peer review and any attached files.

Reviewer #1: No

Reviewer #2: No

---

## [Author Response · Author response to Decision Letter 0]

5 Apr 2022

We have uploaded a detailed letter that responds to the reviewers and editors, showing how all requests were incorportated.

---

## [Decision Letter · Decision Letter 1]

16 May 2022

PONE-D-21-23166R1Exposure to Stereotype-Relevant Stories Shapes Children’s Implicit Gender StereotypesPLOS ONE

Dear Dr. Block,

Thank you for submitting your manuscript to PLOS ONE. After careful consideration, we feel that it has merit but does not fully meet PLOS ONE’s publication criteria as it currently stands. Therefore, we invite you to submit a revised version of the manuscript that addresses the points raised during the review process. The reviewers and I greatly appreciate how thoroughly you addressed their concerns and suggestions. The manuscript is much improved and makes a nice contribution to the literature. The reviewers have a few minor suggestions to strengthen the paper. In particular, please report all of the results from the regression analysis with age as a continuous variable in the supporting materials (see Reviewer 1) and elaborate on issues related to the age median split (see Reviewer 2). Once these points are addressed, I am happy to accept the manuscript. 

We look forward to receiving your revised manuscript.

Kind regards,

Natalie J. Shook

Academic Editor

PLOS ONE

Journal Requirements:

Reviewers' comments:

Reviewer's Responses to Questions

**Comments to the Author**

1. If the authors have adequately addressed your comments raised in a previous round of review and you feel that this manuscript is now acceptable for publication, you may indicate that here to bypass the “Comments to the Author” section, enter your conflict of interest statement in the “Confidential to Editor” section, and submit your "Accept" recommendation.

Reviewer #1: All comments have been addressed

Reviewer #2: All comments have been addressed

2. Is the manuscript technically sound, and do the data support the conclusions?

Reviewer #1: Yes

Reviewer #2: Yes

3. Has the statistical analysis been performed appropriately and rigorously? 

Reviewer #1: Yes

Reviewer #2: Yes

4. Have the authors made all data underlying the findings in their manuscript fully available?

Reviewer #1: Yes

Reviewer #2: Yes

5. Is the manuscript presented in an intelligible fashion and written in standard English?

Reviewer #1: Yes

Reviewer #2: Yes

6. Review Comments to the Author

Reviewer #1: The authors have done a nice job in addressing the reviewer comments. They have successfully incorporated discussion of the limitation of non-random assignment and have nicely discussed implications of all results, including those related to math self-concept.

I noticed some incongruencies in how the participant age X condition analyses are reported across the main-text and supplement. That made it difficult to compare the continuous results against the categorical results. My only outstanding recommendation would be for the authors to synchronize the statistical reporting across the main-text and supplement. Specifically, I recommend reporting the following additional results to the supplemental analysis:

1. The main effect of age

2. Simple slope analyses to elaborate on the interaction effect (e.g., +/-1 SD on age)

Reviewer #2: Thank you for the opportunity to review this paper – I was an original reviewer on the manuscript. The original strengths of the paper are retained. The authors addressed my concerns about the introduction, discussion, and method by adding citations, using more careful language, changing the framing of the implicit stereotype change question, and clarifying the demographic data source. The counterarguments presented in explaining why some of my suggestions were not incorporated make perfect sense, and I appreciate the thorough and thoughtful response to each suggestion.

I looked carefully at how the median split results compared to the regression results. I think the addition to p. 9, setting up age as a categorical vs. continuous construct, is useful for setting up that analytic choice. I still think it is a bit unusual to use median split to distinguish between qualitative categories – if the group had been slightly older on average, the argument would be that a 9.5 year old is qualitatively different than a 9.6 year old, but in this case, the argument is that a 8.9 year old is qualitatively different than an 8.8 year old. In other words, median splits produce groups deemed qualitatively different based on the particular age range of that particular sample, rather than based on broader developmental trajectories. I might make that clearer in the document when comparing this median split to Gonzalez’ (p. 19 of the tracked changes edition). I think including the regression outcomes in the SI is also a good addition to the paper.

I note some small corrections below.

Minor issues (pages based on the tracked changes edition)

1. P. 8: the new sentence beginning with “Particularly” is a fragment.

2. P. 20, line 368, “important to examine” instead of “important examine”

7. PLOS authors have the option to publish the peer review history of their article (what does this mean?). If published, this will include your full peer review and any attached files.

Reviewer #1: No

Reviewer #2: No

---

## [Author Response · Author response to Decision Letter 1]

13 Jun 2022

The response to reviewers was uploaded for better formatting:

Dear Dr. Shook,

Thank you for your help and quick turn around on this paper. We really appreciate the opportunity to publish our work (PONE-D-21-23166R1) at PLOSone. We thank you and the reviewers for taking another look at our manuscript and helping us improve the manuscript before publication. As detailed below, we have carefully addressed comments made by the reviewers in both the main manuscript and the supplement. With these changes, we believe we were able to address all concerns raised by the reviewers.

There are no changes to the financial disclosure we have to report. We have also uploaded our protocols and de-identified data under https://osf.io/2h3xu/.

Thank you for your time,

Katharina Block (on behalf of Andrew Baron, Antonya Gonzalez, Clement Choi, Zoey Wong, and Toni Schmader)

Editor Comments:

The reviewers and I greatly appreciate how thoroughly you addressed their concerns and suggestions. The manuscript is much improved and makes a nice contribution to the literature. The reviewers have a few minor suggestions to strengthen the paper. In particular, please report all of the results from the regression analysis with age as a continuous variable in the supporting materials (see Reviewer 1) and elaborate on issues related to the age median split (see Reviewer 2). Once these points are addressed, I am happy to accept the manuscript. 

Comments to the Author

1. If the authors have adequately addressed your comments raised in a previous round of review …?

Reviewer #1: All comments have been addressed

Reviewer #2: All comments have been addressed

2. Is the manuscript technically sound, and do the data support the conclusions?

Reviewer #1: Yes

Reviewer #2: Yes

3. Has the statistical analysis been performed appropriately and rigorously?

Reviewer #1: Yes

Reviewer #2: Yes

4. Have the authors made all data underlying the findings in their manuscript fully available?

Reviewer #1: Yes

Reviewer #2: Yes

5. Is the manuscript presented in an intelligible fashion and written in standard English?

Reviewer #1: Yes

Reviewer #2: Yes

6. Review Comments to the Author

Reviewer #1: 

A. The authors have done a nice job in addressing the reviewer comments. They have successfully incorporated discussion of the limitation of non-random assignment and have nicely discussed implications of all results, including those related to math self-concept.

Author Response: 

Thank you for your time and your help with improving these parts of the manuscript. We are happy to hear that you are satisfied by our improvements.

B. I noticed some incongruencies in how the participant age X condition analyses are reported across the main-text and supplement. That made it difficult to compare the continuous results against the categorical results. My only outstanding recommendation would be for the authors to synchronize the statistical reporting across the main-text and supplement. Specifically, I recommend reporting the following additional results to the supplemental analysis:

1. The main effect of age

2. Simple slope analyses to elaborate on the interaction effect (e.g., +/-1 SD on age)

Author Response: 

Thank you for these suggestions. Upon reading these sections side by side, we agree that make the main manuscript and the supplement for congruent would help the reader compared the results. As such, we have added the requested analyses and tried to format the results more in in line with the main manuscript. You can now find these analyses on p. 14-15 of the supplement (in tracked changes version).

Reviewer #2: 

A. Thank you for the opportunity to review this paper – I was an original reviewer on the manuscript. The original strengths of the paper are retained. The authors addressed my concerns about the introduction, discussion, and method by adding citations, using more careful language, changing the framing of the implicit stereotype change question, and clarifying the demographic data source. The counterarguments presented in explaining why some of my suggestions were not incorporated make perfect sense, and I appreciate the thorough and thoughtful response to each suggestion.

Author Response: 

Thank you for your gracious response. We really appreciate the thought and time you have put into helping us making our manuscript better.

B. I looked carefully at how the median split results compared to the regression results. I think the addition to p. 9, setting up age as a categorical vs. continuous construct, is useful for setting up that analytic choice. I still think it is a bit unusual to use median split to distinguish between qualitative categories – if the group had been slightly older on average, the argument would be that a 9.5 year old is qualitatively different than a 9.6 year old, but in this case, the argument is that a 8.9 year old is qualitatively different than an 8.8 year old. In other words, median splits produce groups deemed qualitatively different based on the particular age range of that particular sample, rather than based on broader developmental trajectories. I might make that clearer in the document when comparing this median split to Gonzalez’ (p. 19 of the tracked changes edition). I think including the regression outcomes in the SI is also a good addition to the paper.

Author Response: 

We want to thank you again for your original comments as they have markedly improved our manuscript. We are happy to see that our addition on page 9 feel like an improvement to you. We have now added additional detail comparing the media split to Gonzalez et al. (2017) and additionally added citations for several related papers with similar splits. As we argue, there is a possibility that there is indeed a qualitative difference between age groups. However, we do not mean to suggest we provide strong evidence for this (vs. continuous change) yet. We acknowledge that future research has to be done to actually determine where exactly the shift take place on average (e.g., is the shift between 8 and 9 or 8.5 to 9.5 on average). Thus, we a) added a more detailed mention of analyses with age as a continuous variable (p. 19 of tracked version), and b) mention in the discussion the nature of changes with age need to be better understood with future research (p. 22 of tracked document).

C. I note some small corrections below.

a. P. 8: the new sentence beginning with “Particularly” is a fragment.

Author Response: 

Thank you for noticing that this sentence was unclear. It now reads: “Particularly, gender stereotypes merit separate investigation because children’s understanding of gender and race show different developmental trajectories [28]” 

b. P. 20, line 368, “important to examine” instead of “important examine”

Author Response: 

Thank you for noticing this mistake, we have fixed it. Additionally, we have the whole manuscript another round of proofreading.

---

## [Editor Report · Decision Letter 2]

30 Jun 2022

Exposure to Stereotype-Relevant Stories Shapes Children’s Implicit Gender Stereotypes

PONE-D-21-23166R2

Dear Dr. Block,

We’re pleased to inform you that your manuscript has been judged scientifically suitable for publication and will be formally accepted for publication once it meets all outstanding technical requirements.

Kind regards,

Natalie J. Shook

Academic Editor

PLOS ONE
---

## [Editor Report · Acceptance letter]

13 Jul 2022

PONE-D-21-23166R2 

Exposure to Stereotype-Relevant Stories Shapes Children’s Implicit Gender Stereotypes 

Dear Dr. Block:

I'm pleased to inform you that your manuscript has been deemed suitable for publication in PLOS ONE. Congratulations! Your manuscript is now with our production department. 

Kind regards, 

on behalf of

Dr. Natalie J. Shook 

Academic Editor

PLOS ONE